# The Impact of Regional Integration Strategies on the Formation of City Regions and Its Agglomeration Shadow: Evidence from the Yangtze River Delta, China

**Yanlin Zhen [1], Dehao Shi [2,\*] and Yanan Lu [3,\*]**

1   Spatial Planning Center, Yangtze Delta Region Institute of Tsinghua University, Jiaxing 314006, China;
    zhenyl03@163.com
2   Department of Geography and Resource Management, The Chinese University of Hong Kong,
    Hong Kong SAR 999077, China
3   School of Public Affairs, Zhejiang University, Hangzhou 310030, China
\*   Correspondence: dehao.shi@link.cuhk.edu.hk (D.S.); luyanan@zju.edu.cn (Y.L.)

**Abstract:** Using a sample of 122 county-level units in the Yangtze River Delta (YRD) region from 2000 to 2017, this study employs a difference-in-differences model (DID) to examine the impact of regional integration strategy (RIS) on city-region formation and a difference-in-difference-in-difference model (DDD) to test whether it has spatial heterogeneity. The results indicate that RIS has a significant positive impact on industrial integration while it also displays obvious industrial heterogeneity and spatial heterogeneity. The results of the present study contribute to the following points: First, the implementation of RIS promotes a balanced layout of the secondary industry in the region, yet the tertiary industry tends to agglomerate towards central cities. Furthermore, we found that RIS has a more significant negative effect on the integration of the secondary industry and tertiary industry in cities adjacent to metropolis. Consequently, RIS magnifies the "agglomeration shadow" within city regions in terms of industrial integration. Last, our in-depth fieldwork on Jiaxing unravels the mechanism of the shadow effect of RIS.

**Keywords:** city region; regional integration strategy; industrial integration; DDD model; Yangtze River Delta





## 1. Introduction

The literature on urban and regional development has extensively explored agglomeration and its external economy [1], while city regions have become a prevalent topic since the rise of new regionalism at the turn of the century [2–4]. As a result, spatial reconstruction in city regions, which refers to the core city and its outer surrounding areas, has brought socioeconomic transformations involving urban sprawl [5,6], industrial upgrading [7], and capital accumulation [8,9]. More importantly, framing urban issues on the regional scale does a favor to clarify the inter-scalar and cross-boundary interactions among cities [10]. Such interactions create a dynamic, interdependent, and self-adaptive network for geographically adjacent and economically connected cities. Examining city regions has led to a novel spatial epistemology, paving the way for new paradigms regarding population mobility, industrial restructuring, land use, and urban governance [9,11–16]. Subsequently, research on city regions has gone beyond classic theories such as growth poles and central places and has inspirited many enlightening theories and topics such as multidimensional proximity, global city regions, borrowed size, and agglomeration shadow [17,18]. Recently, the growing spatial polarization and inequality within city regions have been increasingly recognized worldwide [19,20]. In Europe, for example, the discourse on city regions has been extended to highlight the achievements of major cities in terms of innovation, economic growth, dynamism, and global competitiveness, creating a generally unfavorable

context for peripheral cities [19,21]. Therefore, formulating, evaluating, and reflecting on specific RIS to promote just and balanced growth in all cities while also accounting for their spatial heterogeneity has emerged as a new and growing area of interest in the field of city-regional studies [22,23].

Previous studies have explored the distribution, evolution, and mechanism of urban-based socioeconomic activities on regional scales during the formation of city regions [24]. Most of these studies were established on the basis of classical economic theories. According to classical economic geography, the city region is attributed to differential land rent and the transformation of core-peripheral structure [25]. Conversely, empirical studies based on new economic geography focus more on imperfectly competitive markets, bounded rationality, and increasing returns to scale in city-regional formation [25–27]. However, most of the existing literature focused on market-led, spontaneous redistribution in city regions but overlooked the role of state power. This study attempts to verify whether and to what extent RIS promotes the formulation of city regions by exemplifying one of the most developed city regions in China.

Additionally, existing research on city regions has mainly focused on the transformation of economic externalities resulting from spatial networking. Compared to the economic externalities of hierarchical urban systems stemming from physical geographic proximity, networked city regions have been observed across non-adjacent physical spaces. Some scholars conceptualize these externalities as "externality fields" [1], "cluster economies" [28], "complex economies" [29], or "urban network externalities" [30]. Although most current research focuses on conceptualizing and measuring the economic externalities in the process of city-regional formation [31,32], few studies paid attention to whether there is spatial heterogeneity in the externalities of city regions and to what a certain extent. In summary, seminal works have examined the various spatial patterns, mechanisms, and heterogeneity of city regions in areas such as migration, industrial redistribution, and urban governance, producing valuable insights and conclusions. However, to date, there is no relevant literature that directly investigates the impact of state power on the formation of city regions and their spatial heterogeneity.

Since its accession to the World Trade Organization in 2001, China has increasingly become a magnet for multinational corporations and capital, thereby necessitating a recalibration of its regional strategy to better align with market-oriented accumulation and governance [14,33]. Subsequently, the Chinese government gradually embraced the concept of new regionalism and presented their interpretation of the city region, referred to as the metropolitan area. These are regional entities with a high-level city at the core, such as the Beijing metropolitan area, Shanghai metropolitan area, Guangdong metropolitan area, and others. Following the 2008 financial crisis, Chinese cities faced evident surplus capacity and regulatory crises. To address these challenges, China initiated adjustments to the previous city-regional model that centered around a single city and shifted towards a polycentric approach to city regions, namely urban agglomeration. In contrast to the city regions driven by market forces under Western neoliberalism since the late 20th century, China's city regions exhibit a more pronounced state intervention [33,34]. To be specific, to accelerate the cultivation of city regions, the Chinese governments formulated targeted regional integration strategies (RIS), aiming to reduce institutional barriers and accelerate the cross-border flow of resources, thereby enhancing the economic interaction and industrial division of labor among cities within specific regions.

Thereupon, we deduce that the mushrooming RIS in China and the rapidly urbanizing hinterland within city-region formation provide a fertile ground for investigating this topic. This raises several questions that stimulate our interest: (1) Does the state-guided strategy of regional integration, which functions as an essential symbol of city-region formation, advance the formation of city regions, and to what extent? (2) What kinds of heterogeneity are existed in such state-orchestrating city regions, particularly in terms of industrial and spatial structure?

This study focuses on the Yangtze River Delta, one of the most advanced and net-worked city regions in China, including all county-level units in Shanghai Municipality, Zhejiang Province, and Jiangsu Province. The "Yangtze River Delta Regional Plan" (YRDRP), approved by the Chinese central government in 2010, was regarded as a quasi-natural experiment in this study. Based on panel data on secondary and tertiary sectors of 121 urban entities in the YRD from 2000 to 2017, we use a difference-in-differences model (DID) to empirically estimate the net effect of RIS on the city region construction. Additionally, a difference-in-difference-in-difference model (DDD) was employed to further identify whether the formation of city regions has potential spatial heterogeneity in inter-city interactions (i.e., agglomeration shadow).

## 2. Literature Review and Research Hypotheses

### 2.1. City-Regional Integration and Regional Integration Strategy

The emergence of city regions, spanning from megalopolis to urban agglomeration, has attracted wide attention from scholars of geography, economy, and political studies. The formation of city regions entails multiple regional integrations involving economic, industrial, institutional, and cultural integration, which are regarded as essential agendas. Currently, explanations of the spatial dynamics of city-regional integration mainly focus on two aspects. Some literature pays attention to the socioeconomic effect of city regions, suggesting that the mechanism of RIS promoting city-regional formation lies in encouraging and accelerating cross-border and cross-boundary flowing of labor, capital, and other factors [10,14]. Another line of research places more emphasis on the state agenda and highlights that the formation of city regions is a response to tensions surrounding social reproduction and sustainability within the city region. It is guided by the government's adoption of the new regionalism approach [33,35]. The transformation of state power in terms of reterritorialization and rescaling in capital accumulation provides an insightful understanding of the spatial reorganization of city-regional integration [36]. The former approach emphasizes the quantitative analysis of inter-city interactions in the formation of city regions, while the latter focuses on elucidating the policy utility of this process.

Despite growing calls for city-region integration to be examined through the lens of multi-agent intermeshing negotiation [11,37] or internalization of externalities brought about by cross-border mobility [38], there is a lack of research that integrates both perspectives. It is unclear whether there is a causal relationship between the implementation of RIS, which represents state intervention, and the movement of socioeconomic activities, which represents inter-city interactions. Yet, no consensus has been reached on this issue. Therefore, this study will focus on whether RIS significantly affects the relocation of socioeconomic activities within city regions.

Generally, city regions consist of central cities that are dominated by tertiary industries, such as producer services and finance, and peripheral regions that are dominated by secondary industries, such as manufacturing and processing industries. The YRDRP in 2010 emphasized economic integration through industries relocation as the primary goal. Specifically, the strategy explicitly stated that the modern service industry needs to concentrate in central metropolises such as Shanghai, Nanjing, and Hangzhou, while the manufacturing industry is envisioned to relocate to other cities within the region to achieve a more balanced spatial distribution. The decentralization of secondary industries and the agglomeration of tertiary industries represent the formation of city regions, as well as industrial integration [39,40]. Thus, this paper proposes the first hypothesis that the implementation of regional integration strategies (RIS) will significantly promote industrial integration in the process of city-region formation. Additionally, we argue that the manufacturing and service industries, which, respectively, dominate the second and third sectors, will display distinct spatial patterns because of RIS implementation.

### 2.2. Spatial Heterogeneity within City Regions

The formation of city regions marks a transition of the regional organization from a hierarchical to a networked structure, signifying a fundamental shift in the paradigm of intercity interactions. Alonso (1973) introduced the concept of "borrowed size" and demonstrated that small cities could benefit from agglomeration economies "borrowed" from larger neighboring cities. Scholars, including Burger, Meijers, Hoogerbrugge and Tresserra [18], Meijers and Burger [41], and Volgmann and Rusche [42], have advanced the measurement of borrowed size and employed it in the process of paradigm shifting from a hierarchical urban system to networked city region. To be specific, in the context of the hierarchical urban system, agglomeration externalities in specific cities are typically rooted in the size of the city, with advanced urban functions, industrial structure, and modernity often being positively correlated with larger city sizes. Nonetheless, as city regions are constructed based on networked spatial interactions, the aforementioned causation has been gradually decoupled [43]. Small and medium-sized cities located within city regions are capable of hosting urban functions that may surpass the population size, owing to drawing upon the population necessary to support these functions from the wider urban network [44]. Such spatial effects, termed "borrowed size", have emerged as an effective strategy for peripheral cities to achieve greater economic development and ascend to higher levels within the context of urban region construction. On the contrary, the concept of "agglomeration shadow" was conceptualized from the opposite spatial effect of "borrowed size", referring to the possibility that residents of peripheral cities work and consume in core cities. Small and medium-sized cities located at the periphery of large urban centers may experience reduced or even counteractive effects resulting from spatial networking.

In recent years, the city-regional characteristics have gradually emerged in the Yangtze River Delta, as well as the agglomeration shadow effect. For example, Jiaxing, which is one of the prefecture-level cities adjacent to Shanghai, has experienced significant agglomeration shadows in many aspects, including population immigration, industrial upgrading, and more. Some studies based on the new regionalism perspective suggest that local governments should propose more powerful integration policies to eliminate the shadow effects. However, few studies have investigated whether and to what extent such shadow effects are rooted in RIS. Subsequently, this paper proposes a second research hypothesis that regional integration strategies will intensify spatial heterogeneity in the process of city-regional formation.

On the basis of the theoretical discussion and research hypotheses, the research framework in this paper is shown in the Figure 1.

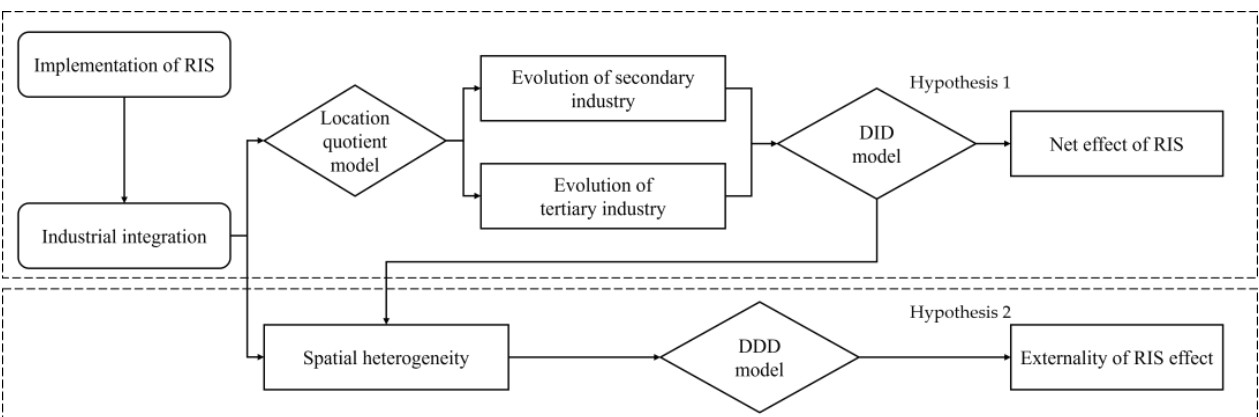

**Figure 1.** Theoretical mechanism of RIS and research framework.

## 3. Methodology and Data Collection

### 3.1. Study Area

According to the YRDRP published in 2010, the initial integration strategy of the Yangtze River Delta primarily focused on Shanghai, Jiangsu Province, and Zhejiang Province. The plan identified 16 cities as the core area, which included the province-level municipality of Shanghai, 8 out of the 13 prefecture-level cities in Jiangsu Province (namely Nanjing, Suzhou, Wuxi, Changzhou, Zhenjiang, Yangzhou, Taizhou, and Nantong), and 7 out of the 11 prefecture-level cities in Zhejiang Province (namely Hangzhou, Ningbo, Huzhou, Jiaxing, Shaoxing, Zhoushan, and Taizhou). In addition, the above-mentioned 15 prefecture-level cities and the Shanghai municipality serve as the core implementation areas of RIS, while county-level units within these areas are labeled as the Treated group of RIS, and county-level units of other cities are noted as the Control group. It should be underlined that the focus of the city-regional study is the interaction of urban entities, so municipal districts within a specific prefecture-level city are regarded as a single statistical unit. A total of 121 urban entity units were included in this study.

To verify the potential heterogeneity of the city region in the second hypothesis, it is necessary to identify research objects where potential shadow effects may occur. Based on existing research and on-site investigations in the Yangtze River Delta region, this study designates county-level units adjacent to the municipal areas of Shanghai, Nanjing, and Hangzhou as potential shadow areas, which are labeled as Shadow groups in the DDD model. All other county-level units are labeled as non-shadow groups (See Figure 2).

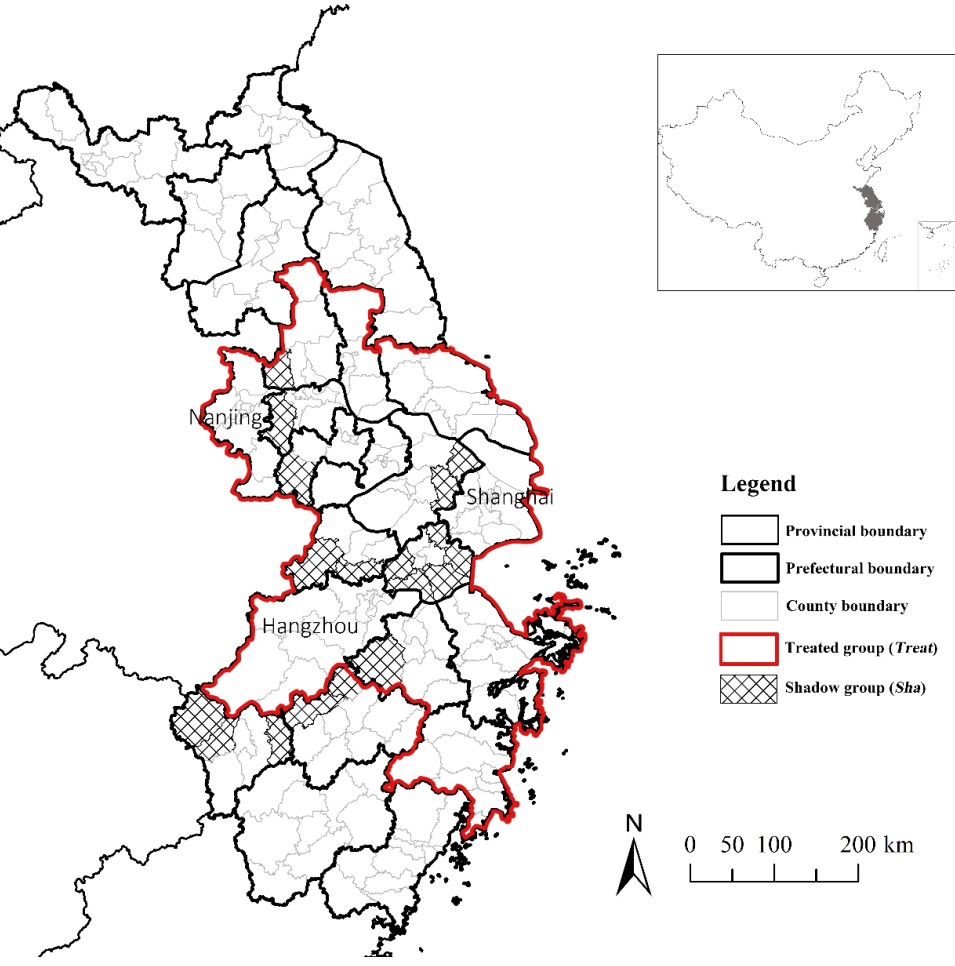

**Figure 2.** The Yangtze River Delta, the treated group of RIS and potential shadow group.

Since the turn of the new millennium, China has increasingly welcomed multinational corporations to establish production departments in domestic, especially after China's accession to the World Trade Organization in 2001. The entry of multinational corporations and the expansion of the manufacturing industry have gradually accelerated intercity interactions in the Yangtze River Delta region. Prior to the commencement of the trade dispute between China and the United States in 2018, foreign investment continued to flow into the Yangtze River Delta area, facilitating the reallocation of industries. Consequently, to encapsulate China's duration of engagement with globalization, the research period of the present study spans from 2000 to 2017.

*3.2. Methodology*

To validate the above two hypotheses, the empirical process of this study follows three steps. First, we attempt to characterize the industrial integration of the Yangtze River Delta region using the location quotient method. The measure of location entropy can provide a more intuitive reflection of the spatial reallocation of specific industries, and it serves as a parameter with concrete meaning for explaining the results of subsequent DID models. Since the social and economic data used in this study are at the county level, which allows for more precise identification of urban entities, industry data can only be classified into the secondary and tertiary sectors rather than specific industrial categories, which is one of the limitations of this study. Subsequently, as the implementation of RIS can be viewed as a quasi-natural experiment that has been extensively scrutinized using the DID model, we would utilize the DID model as the baseline regression method to assess the net impact of RIS on industrial integration. Finally, the DDD model would be employed to verify the RIS's spatial heterogeneity on industrial integration. This step would determine whether there is an "agglomeration shadow" or "borrowed size" caused by RIS.

3.2.1. Location Quotient Model

This paper utilizes the location quotient method to calculate the degree of regional industrial integration. If the location quotient of a certain industry in a specific region increases, it suggests that the region has a higher specialization in that industry. The location quotient model is constructed as follows:

$$\gamma_{it} = \frac{e_{ir}}{\sum_i e_{ir}} \bigg/ \frac{\sum_r e_{ir}}{\sum_i \sum_r e_{ir}} \tag{1}$$

where variable $\gamma_{it}$ denotes the index of industrial integration of industry $r$ in city $i$. Variables LQSI and LQTI refer to the result of location quotient analysis conducted on the secondary and tertiary industry, respectively. LQSI and LQTI denote the ratio of industry proportion in a given area to that in the entire region. A higher coefficient of this variable implies a greater concentration of the industry within the city region, leading to both a comparative and a relative scale advantage.

3.2.2. DID Model

This paper uses the RIS in the YRD implemented in 2010 as a quasi-natural experiment. The DID model is constructed as follows:

$$\gamma_{it} = \beta_0 + \beta_1 RIS_{it} + \beta_2 Treat_i + \beta_3 T_t + \tau Cov_{it} + \mu_i + \delta_t + \varepsilon_{it} \tag{2}$$

where variable $\gamma_{it}$ denotes the index of industrial integration (the result of location quotient analysis) for city $i$ in year $t$. Referencing from the existing literature about the city-regional industrial effect, we take the output changes of the secondary and tertiary industries as the dependent variables. $Treat_i$ is a dummy variable, referring to whether the city $i$ belongs to the sixteen-core prefecture-level cities of the YRD. If city $i$ is part of core region, $Treat_i = 1$; otherwise, $Treat_i = 0$. Similarly, $T_t$ is a time dummy variable, showing that quasi-natural experiment was implemented at year $t$. The YRDRP was promulgated in 2010; thus, $T_t = 1$

after 2010; otherwise, $T_t$= 0 before 2010. $RIS_{it}$ is the most important variable representing the RIS effect, which is equal to $Treat_i \times T_t$. The estimated $\beta_1$ captures the effect of RIS implemented in a city on industrial integration. If $\beta_1$ is significantly positive, it means that the implementation of RIS improves the industrial specialization within the city region. On the contrary, the significantly negative $\beta_1$ indicates a homogenization trend caused by RIS.

In addition, the parameters $\beta_2$ and $\beta_3$ represent the difference in the RIS effect between the presence and absence and after and before, respectively. $Cov_{it}$ reflects a set of control variables, which is not fixed in this study due to the influence of different factors on the industrial integration of secondary and tertiary industries. Variables $\beta_0$ and $\varepsilon_{it}$ indicate the constant and residual items, respectively. The variable $\mu_i$ is the city fixed effect used to control urban heterogeneity, and $\delta_t$ is the time fixed effect used to control the time trend.

### 3.2.3. DDD Model

In order to estimate the heterogeneity of the effect across the industrial evolution of RIS, this paper proposes the following DDD model by adding the variable *Sha* and its interaction term with $RIS_{it}$, $Treat_i$ and $T_t$ into DID model:

$$\gamma_{it} = \alpha_0 + \alpha_1 RIS_{it} \times Sha_i + \alpha_2 Sha_i \times Treat_i + \alpha_3 Sha_i \times T_t + \alpha_4 RIS_{it} + \alpha_4 Sha_i \\ + \alpha_5 Treat_i + \alpha_6 T_t + \tau Cov_{it} + \mu_i + \delta_t + \varepsilon_{it} \tag{3}$$

where $Sha_i$ denotes a dummy variable indicating whether city $i$ is potentially affected by the shadow effect. The estimated $\alpha_1$ captures the heterogeneity of the effect of RIS. If city i is located in the vicinity of the metropolis areas (Shanghai, Hangzhou, and Nanjing) of YRD, $Sha_i = 1$, whereas if not, $Sha_i$= 0. A significant positive value of $\alpha_1$ indicates that the implementation of RIS brings a "borrowed size" effect to county-level units adjacent to the municipal areas. Otherwise, a significantly negative value of $\alpha_1$ means that the implementation of RIS brings an "agglomeration shadow" effect.

### 3.3. Data Source and Descriptive Statistics of Variables

The data sources used in this study include statistical yearbooks from Zhejiang Province, Jiangsu Province, and Shanghai Municipality for information on the gross output value of the secondary industry, the gross output value of the tertiary industry, and per capita GDP. Meanwhile, data on fixed asset investment, real estate investment, public fiscal expenditure, and patents granted for the invention are gleaned from the statistical yearbooks of diverse prefecture-level cities and counties. Given the difference of the units of measurement among the variables, the data for variables, including secondary industry, tertiary industry, population number, gross domestic product per capita, fixed-asset investment, real estate investment, public financial expenditure, granted patents, and industrial structure are processed into logarithmic form in the subsequent analysis. Due to the constraints of county-level statistical data and the need for a sufficiently long panel data period for the DID model, we are only able to describe the industrial structure using basic output values of the secondary and tertiary industries rather than more detailed industrial categories.

As aforementioned, the indication of regional integration and city-regional formation in the industrial dimension are the tendencies for the tertiary industry to be concentrated and specialized while the secondary sector to be balanced. Thus, both the location quotient index of secondary industry decreasing and that of tertiary industry increasing indicated a positive tendency of region integration. The descriptive statistical results of the above variables are shown in the Appendix A. The LEST and LETI of cities range from 0.26 to 1.36 and 0.60 to 1.52, respectively, which indicated that the degree of agglomeration and specialization of the secondary industry is lower than that of the tertiary industry.

## 4. Regression Results

### 4.1. Benchmark Regression

First, we used the location quotient to measure the industrial integration change of secondary and tertiary sectors in the YRD during the study period. Furthermore, the promotion effect of RIS on the formation of city regions is estimated by benchmark regression, as shown in Table 1. The corresponding columns (1) and (2) list the variables and coefficients of the integration of secondary and tertiary industries. All parameters are estimated using robust standard errors.

**Table 1.** Benchmark regression results.

| Variables | (1) LQSI | Variables | (2) LQTI |
|---|---|---|---|
| RIS | −0.016 *** | RIS | 0.002 * |
| | (−6.31) | | (0.71) |
| GDP | −0.072 *** | POP | −0.059 *** |
| | (−14.38) | | (−8.63) |
| pGDP | 0.010 *** | pGDP | 0.027 *** |
| | (23.28) | | (5.02) |
| PFE | 0.020 *** | PFE | 0.009 *** |
| | (4.23) | | (1.41) |
| FAI | 0.004 ** | PAT | −0.005 * |
| | (2.43) | | (−3.5) |
| RSI | 0.003 *** | RSI | 0.387 *** |
| | (2.99) | | (82.29) |
| INDS | −0.446 *** | INDS | 0.001 *** |
| | (−112.82) | | (0.32) |
| _cons | 0.060 * | _cons | 1.00585 *** |
| | (1.81) | | (17.23) |
| City FE | Yes | City FE | Yes |
| Year FE | Yes | Year FE | Yes |
| N | 2178 | N | 2178 |
| R_square | 0.98 | R_square | 0.98 |

Notes: The t-statistics for the coefficients are reported in parentheses. Symbols *, **, and *** denote significance levels of 0.1, 0.05, and 0.01, respectively. Column (1) and (2) present coefficients of the integration of secondary and tertiary industries, respectively.

The results show a significant negative coefficient of RIS in LQSI, indicating that the spatial distribution of the secondary industry is increasingly balanced. On the contrary, the significant positive coefficient of RIS indicates that the tertiary industry is more polarized since the implementation of the integration policy. These results demonstrate that the Yangtze River Delta region has displayed a substantial trend of industrial integration since the implementation of RIS. The benchmark regression can reveal a correlation between RIS and industrial integration, but more evidence is required to establish their causative connection.

### 4.2. RIS on Industrial Integration

Using the DID model, we conducted an estimation to determine the total impact of RIS on industrial integration. The results are presented in Table 2. Specifically, the first and second columns of the table display the net effect of RIS on the second and third industries, respectively. The first column shows an increasing trend in the range of LQSI coefficients, rising from a range of −1.156~−1.152 to −1.129~−1.107. This indicates that there is an agglomeration of the secondary industry in the YRD, which poses challenges to regional integration. Moreover, the coefficient for RIS, at 0.019, is significantly negative. This indicates that RIS will lead to a reduction in the agglomeration trend of the secondary industry in the YRD.

**Table 2.** The effect of RIS on industrial integration.

| Variables | | (1) LQSI | Variables | | (2) LQTI |
|---|---|---|---|---|---|
| Before | Control | −0.152 | Before | Control | 1.360 |
| | Treated | −0.156 | | Treated | 1.353 |
| | Diff(T-C) | −0.004 (−0.72) | | Diff(T-C) | −0.007 (−0.840) |
| After | Control | −0.107 | After | Control | 1.233 |
| | Treated | −0.129 | | Treated | 1.250 |
| | Diff(T-C) | −0.022 *** (4.48) | | Diff(T-C) | 0.016 *** (2.91) |
| RIS | | −0.019 *** (2.86) | RIS | | 0.024 *** (2.49) |
| GDP | | 0.005 * (1.143) | POP | | −0.003 (−0.472) |
| pGDP | | 0.097 *** (22.636) | pGDP | | −0.034 *** (−3.404) |
| PFE | | −0.028 *** (−5.279) | PFE | | −0.036 *** (−4.772) |
| FAI | | 0.003 * (0.755) | PAT | | 0.020 *** (8.931) |
| RSI | | 0.015 *** (5.805) | RSI | | 0.036 *** (10.832) |
| INDS | | −0.400 *** (−63.048) | INDS | | 0.425 *** (38.572) |
| N | | 2178 | N | | 2178 |
| R_square | | 0.85 | R_square | | 0.79 |

Notes: The t-statistics for the coefficients are reported in parentheses. Symbols *, *** denote significance levels of 0.1, 0.01, respectively. Column (1) and (2) present coefficients of the integration of secondary and tertiary industries, respectively.

The second column displays the net effect of RIS on the tertiary industry. The coefficients for the tertiary industry are positive, indicating a growing trend towards concentration and specialization in the tertiary industry in the YRD. However, the result indicated that such a trend is converging as the range of the LTSI coefficients decreases from 1.353~1.360 to 1.233~1.250. However, the coefficient for RIS is 0.024, which is significantly positive, suggesting that RIS policies are contributing to the increasing specialization of the tertiary industry in the YRD. In sum, the implementation of RIS has a noticeable positive impact on industrial integration, as it facilitates the spread of the secondary industry and the concentration of the tertiary industry.

The impact of RIS on the secondary and tertiary industries runs counter to the spontaneous evolution trends of the industries. To be specific, entrepreneurial local governments in the YRD have demonstrated intense intercity competition and industry homogenization in order to attract external investments and manufacturing firms [45,46]. As a result of their more sophisticated infrastructure and robust financial capabilities, developed cities have been more successful in luring investments, which has led to a persistent trend of secondary industry concentration and increasing economies of scale. Notably, the coefficient of LQSI is significantly negative, indicating that RIS can play a pivotal role in promoting a more balanced spatial distribution of the secondary industry and mitigating deleterious intercity competition. Li and Wu [47], for instance, suggested that the YRDYP has employed city-regionalism and promoted cross-border collaboration, and turned intercity competition into intercity cooperation effectively.

On the contrary, it is observed that the tertiary sector is primarily concentrated in metropolitan areas such as Shanghai's financial industry, Hangzhou's internet industry, and Nanjing's producer services industry. During the experimental period, the location quotient of the tertiary industry displayed a notable decrease, thereby signaling the spillover effects of the tertiary industry. In the years following 2010, the agglomeration degree of the tertiary industry recorded a significant increase of 2.4%, which indicates that the RIS played a role

in the concentration of the tertiary industry towards central cities. Scholars in the field of new economic geography have attributed this trend to the increasing returns to scale effect that emerged following the degree of trade freedom [48]. Furthermore, some studies suggest that this is rooted in the tertiary sector, particularly in the financial and internet sectors that require elevated urban platforms to acquire greater financing [49]. In summary, RIS in the YRD has indeed facilitated the establishment of city regions through the lens of industrial structure transformation.

### 4.3. Heterogeneity of RIS on Industrial Integration

To verify whether there are externalities, such as agglomeration shadow and borrowed size effects, in the city-regional formation, this paper further uses the DDD model to examine the spatial heterogeneity of the effect of RIS on industrial integration. The results of the DDD model are presented in Table 3, where the first and second columns of the distribution represent the spatial heterogeneity effects of RIS on industrial isomorphism in the secondary and tertiary industries, respectively. The results show that the coefficients of the interaction term between regional integration RIS and dummy variable Sha before the RIS implementation, 0.013 in LQSI and −0.001 in LQTI, respectively, are statistically insignificant. This indicates that there is no significant economic externality caused by regional integration in the YRD before RIS implementation. However, in the period following RIS implementation, the coefficients of LQSI and LQTI show statistical significance. Thus, the comparison of the significance before and after implementation indicates that RIS's industrial integration effect exhibits spatial heterogeneity in the small and medium-sized cities located in the vicinity of the metropolis.

In Table 3, the interaction term's ($RIS \times Sha$) coefficient in column (1) is negative (−0.035) and statistically significant at the 1% level, revealing a significant decrease in the location quotient of the secondary industry in cities that are contiguous to major metropolitan regions. The results also reveal that the coefficient of LQSI in the DID model is negative (−0.019), which is smaller in magnitude than that in the DDD model. These findings suggest that RIS's integration effect on the secondary industry is more apparent in the cities bordering major metropolises compared to non-bordering cities. This trend may be attributed to enhanced intercity transportation, resulting in spillover effects on land prices in cities adjacent to a metropolis, which disincentivizes manufacturing industries to relocate to these cities. Moreover, it is plausible that RIS has facilitated improved accessibility to transportation throughout the entirety of the YRD, thereby permitting peripheral cities previously hampered by transportation disadvantages to gradually emerge as destinations for relocated manufacturing industries.

In terms of the tertiary sector, the coefficient of LQTI is significantly negative (−0.048), indicating that the advantage degree of the tertiary industry of cities bordering major metropolises began to decline after the implementation of RIS. The coefficient of LQTI in the DID model evinces a noteworthy positive impact (0.024), suggesting that in the context of overall RIS-induced augmentation of the third industry location entropy, the third industry edge of the neighboring cities of major metropolitan areas has actually experienced a reduction. Our investigation in Jiaxing highlights the formidable challenge of fostering the growth of the third industry in the interstitial areas between Shanghai and Hangzhou. A significant impediment is the dearth of high-value-added technology enterprises willing to establish themselves in Jiaxing due to the unavailability of suitably skilled human capital in the area. For example, R&D institutions in Jiaxing, including Jiaxing Science and Technology City, XiuZhou National High-tech Zone, and the Yangtze River Delta Research Institute, all confront talent shortages and limited expansion capacity. On the other hand, the proximity of Jiaxing to metropolitan areas results in a considerable proportion of the local populace's consumption activities being siphoned off by the neighboring metropolises. Notably, residents in Tongxiang, Haining, and Haiyan counties tend to travel to Hangzhou for weekend consumption, while those in Pinghu and Jiashan counties tend to spend their savings in Shanghai.

**Table 3.** Spatial heterogeneity of RIS on the industrial integration.

| | Variables | (1) LQSI | | Variables | (2) LQTI |
|---|---|---|---|---|---|
| Before | Control & Shadow | −0.122 | Before | Control & Shadow | 1.336 |
| | Control & n-shadow | −0.129 | | Control & n-shadow | 1.359 |
| | Treated & Shadow | −0.114 | | Treated & Shadow | 1.328 |
| | Treated & n-shadow | −0.134 | | Treated & n-shadow | 1.352 |
| | Diff(T-C) | 0.013 (1.31) | | Diff(T-C) | −0.001 (0.08) |
| After | Control & Shadow | −0.043 | After | Control & Shadow | 1.275 |
| | Control & n-shadow | −0.088 | | Control & n-shadow | 1.226 |
| | Treated & Shadow | −0.086 | | Treated & Shadow | 1.246 |
| | Treated & n-shadow | −0.109 | | Treated & n-shadow | 1.246 |
| | Diff(T-C) | −0.022 ** (2.02) | | Diff(T-C) | −0.049 *** (4.14) |
| $RIS \times Sha$ | | −0.035 *** (−2.39) | $RIS \times Sha$ | | −0.048 *** (−2.78) |
| GDP | | 0.005 (1.151) | POP | | −0.002 (−0.269) |
| pGDP | | 0.094 *** (21.848) | pGDP | | −0.034 *** (−3.382) |
| PFE | | −0.025 *** (−4.717) | PFE | | −0.038 *** (−5.01) |
| FAI | | 0.003 (0.623) | PAT | | 0.037 *** (10.912) |
| RSI | | 0.015 *** (5.821) | RSI | | 0.020 *** (8.836) |
| INDS | | −0.396 *** (−62.693) | INDS | | 0.425 *** (38.805) |
| N | | 2178 | N | | 2178 |
| R_square | | 0.85 | R_square | | 0.79 |

Notes: The t-statistics for the coefficients are reported in parentheses. Symbols ** and *** denote significance levels of 0.05, and 0.01, respectively. Column (1) and (2) present coefficients of the integration of secondary and tertiary industries, respectively.

To summarize, city regions of the YRD facilitated by RIS have a positive impact on the agglomeration shadow effect of the tertiary industry, resulting in a decline in the concentration degree and relative advantages of the tertiary industry in the peripheral cities of metropolitan regions. This effect is more intricate in the secondary industry. The regression results indicate that the integration effect of the secondary industry induced by RIS is more conspicuous in the periphery of metropolitan areas, exhibiting the traits of "borrowed size". This entails that the spillover of purchasing power from the metropolitan area has inflated land prices, hastening the exit of the secondary industry. Simultaneously, as this benefit compounds the withdrawal of the secondary industry, it gives rise to a slump in the secondary industry in the vicinity of the metropolitan area, which can be regarded as an "agglomeration shadow". Consequently, this paper concurs with Meijers and Burger's [41] notion that the fundamental mechanism of "borrowed size" and "agglomeration shadow" is identical.

### 4.4. Robustness Checks

To ensure the reliability and credibility of our findings, we have performed a parallel trend test, which validates the significant influence of RIS on industrial integration. One of the pre-conditions for employing the DID model and the DDD model is that the treatment group and control groups should have the same trends before the quasi-natural experiment, as verified through the parallel trend test. The results of the parallel trend test are presented in Table 4, where the dummy variable *Post* refers to the year 2010. The coefficients of '*RIS × Before n*' represents the change in the dependent variables in the *n*th year before the implementation of RIS, while the '*RIS × After n*' represent that in the *n*th year after. As

shown in Table 4, the coefficients of the '*RIS × Before n*' term are statistically insignificant. Instead, after the implementation of RIS, the parameter becomes statistically significant. The observed result aligns with the assumption of parallel trends, which offers additional evidence to support the empirical identification. Furthermore, Table 4 demonstrates that there is a time lag in the impact of RIS on industrial integration. The changes in the LQSI and LQTI did not show statistical significance until the second year after the implementation of RIS.

**Table 4.** Results of the parallel trend test.

| Variables | LQSI | LQTI |
|:---:|:---:|:---:|
| *RIS × Before* 2 | 0.001 | 0.001 |
| | −0.03 | −0.12 |
| *RIS × Before* 1 | 0.002 | 0.007 * |
| | −0.5 | −1.24 |
| *RIS × Post* | −0.011 ** | −0.010 ** |
| | (−2.24) | (−1.39) |
| *RIS × After* 1 | 0.002 | −0.001 |
| | −0.57 | (−0.04) |
| *RIS × After* 2 | −0.014 *** | 0.011 *** |
| | (−3.43) | −3.16 |
| Constant | −0.027 | 0.439 ** |
| | (−0.25) | −2.25 |
| City FE | Yes | Yes |
| Year FE | Yes | Yes |

Notes: The t-statistics for the coefficients are reported in parentheses. Symbols *, **, and *** denote significance levels of 0.1, 0.05, and 0.01, respectively.

## 5. Discussion

This study evaluates the net effect of state-led RIS on the dimensions of industrial upgrading in the formation of city regions, which highlights the pivotal role of regional integration strategy in transforming the spatial structure of city regions through industrial restructuring. These policies have catalyzed transportation infrastructure, accelerating socioeconomic activities transfer and flow within the region. Consequently, this has led to spatial restructuring with more intense flows of people, goods, and information, facilitating the transition from hierarchical to networked spatial structures [50]. The results provide evidence of the policy's impact on spatial structural changes [16], thus providing a valuable reference point for other regions globally that are seeking to promote the formation of city regions and effect industrial restructuring. However, the hypothesis posited by this study is predicated on the impact of state-led strategies and their top-down effect on city regions. The potential presence of bottom-up influences necessitates further exploration [32].

Moreover, this article takes into account the heterogeneity of the scale of city-regional formation in different regions. The reform of "province-directly-administered-county" in the YRD has granted county-level governments greater autonomy in economic development and urbanization [51]. Previous spatial statistical studies on city regions have predominantly posited prefecture-level cities [52], yet these cities are often composed of multiple urban entities. In contrast, county-level units in China exhibit a closer resemblance to a single physical city, with distinct boundaries of the built environment, independent administrative power, and self-contained infrastructure systems. The YRD is characterized by county-based industrial distribution, such as Yiwu in Jinhua City, Zhejiang Province, and Kunshan in Suzhou City, Jiangsu Province. Therefore, using county-level data can provide a more detailed description of the spatial effects of RIS in the YRD and enable a more precise reflection of the process and mechanism of city region formation [52]. Nevertheless, this is not to reiterate that all similar studies should adopt the county-level scale but rather to emphasize that the selection should be based on the socio-economic conditions of the specific study area. In addition, the limitations of the available data precluded a direct

assessment of the interplay among county-level units. As such, this study represents a preliminary exploration that underscores the need for future research to bridge this gap.

## 6. Conclusions

This paper examines the impact of the RIS on the formation of city-region and spatial heterogeneity in the Yangtze River Delta, China. Using data from 121 urban entities, this study analyzed the effects of RIS on industrial integration through a DID model and evaluated spatial heterogeneity in potential shadow areas around the metropolis through a DDD model.

Based on the DID model, this study unveiled the heterogeneous impact of RIS in the YRD on different industrial sectors, whereby high value-added tertiary industries tend to agglomerate in central cities while secondary industries spread to peripheral regions. As a result, central cities emerge as hubs dominated by "headquarters" and "producer services", while peripheral regions are typified by "branches" and "manufacturing". This contributes to the literature about the spatial pattern and heterogeneity of city regions in the industrial dimension [12]. In addition, this article has provided a potential explanation for the causation. It suggests that regional integration policies facilitate factor mobility within the region, leading to a shift from a rent model centered on individual cities to one encompassing the entire region.

Drawing on the DDD model, this study investigated agglomeration shadow effects in city regions. RIS has contributed to tertiary industry concentration in central metropolises, resulting in a substantial decline in peripheral areas. Conversely, while these policies have facilitated secondary industry diffusion from metropolitan areas, high land, and housing prices in peripheral regions have led the industry to locate in more remote regions. This helps to explain the reasons behind the formation of "agglomeration shadows". Through conducting interviews and research in Jiaxing City, this study identifies rapid land price escalation and inadequate talent attraction as key factors behind the "agglomeration shadow" in city regions, which could be further explored as intermediary variables.

On the basis of the principal research findings, we recommend that local governments should meticulously consider the disparities underpinning the developmental substratum of industries, regions, and urban economies when pursuing regional integration. Precisely, it is imperative that the divergent repercussions of RIS on the secondary and tertiary sectors ought to be taken into consideration. Concomitantly, city governments with different developmental orientations towards specific industries should adopt distinct policy frameworks to respond effectively to regional integration. This will enable them to harness the individual strengths of each industry to achieve greater regional development. Finally, considering the specific externalities generated by the formation of city regions, local governments should recognize their individual advantages and the laws of regional integration to purposefully strengthen cooperation with neighboring units.

**Author Contributions:** Conceptualization, Y.Z. and D.S.; methodology, Y.Z., D.S. and Y.L.; software, D.S.; validation, Y.Z., D.S. and Y.L.; formal analysis, Y.Z., D.S. and Y.L.; investigation, Y.Z.; resources, Y.Z.; data curation, D.S. and Y.L.; writing—original draft preparation, Y.Z., D.S. and Y.L.; writing review and editing, Y.Z., D.S. and Y.L.; visualization, Y.Z. and D.S.; supervision, Y.Z., D.S. and Y.L.; project administration, Y.Z.; funding acquisition, Y.Z. All authors have read and agreed to the published version of the manuscript.

**Funding:** This research received no external funding.

**Data Availability Statement:** The data will be available on relevant request from the corresponding author.

**Acknowledgments:** The authors would like to thank all the anonymous reviewers and editors who contributed their time and knowledge to this study.

**Conflicts of Interest:** The authors declare no conflict of interest. The funders had no role in the design of the study; in the collection, analyses, or interpretation of data; in the writing of the manuscript; or in the decision to publish the results.

## Appendix A

**Table A1.** Descriptive statistics of variables.

| Category | Variable | Variable Symbol | Description | Mean | Std. Dev. | Min | Max |
|---|---|---|---|---|---|---|---|
| Dependence Variables | Secondary industry | SecI (hundred million RMB) | Output-value of secondary industry | 259.97 | 466.17 | 2.49 | 4454.87 |
| | Tertiary industry | TerI (hundred million RMB) | Output-value of tertiary industry | 232.24 | 545.51 | 2.39 | 7500.59 |
| | Location quotient of secondary industry | LQSI | Result of location quotient analysis of secondary industry | 0.96 | 0.17 | 0.26 | 1.36 |
| | Location quotient of tertiary industry | LQTI | Result of location quotient analysis of tertiary industry | 0.92 | 0.15 | 0.60 | 1.52 |
| Dummy variable | Cities affected by RIS | Treat | 1 for cities in sixteen core prefecture-level cities, 0 for others | 0.68 | 0.47 | 0.00 | 1.00 |
| | Cities with potential shadow effects | Sha | 1 for cities adjacent to the municipal areas., 0 for others | 0.12 | 0.33 | 0.00 | 1.00 |
| | Time of implementing RIS | T | 1 for years after RIS implementing, 0 for others | 0.44 | 0.50 | 0.00 | 1.00 |
| | RIS | RIS | equal to Core $\times$ T | 0.30 | 0.46 | 0.00 | 1.00 |
| Covariates | Population number | Pop (ten thousand) | Permanent resident population | 93.65 | 76.71 | 7.62 | 680.67 |
| | GDP per capita | pGDP (RMB) | GDP per resident population | 43,483.35 | 36,006.68 | 2900.00 | 218,984.10 |
| | Fixed-asset investment | FAI (Billion RMB) | Fixed-asset investment | 220.94 | 410.60 | 0.32 | 5176.24 |
| | Real estate investment | REI (Billion RMB) | Real estate investment | 61.42 | 176.09 | 0.01 | 2566.90 |
| | Public financial expenditure | PFE (Billion RMB) | Public financial expenditure | 55.13 | 113.50 | 0.94 | 1396.89 |
| | Patents | PAT | The number of granted patents | 2348.27 | 4737.55 | 2.00 | 49,720.00 |
| | Industrial structure | INDS | Tertiary industry divided by secondary industry | 0.82 | 0.33 | 0.32 | 4.49 |

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
