# Peer review of "The Impact of Regional Integration Strategies on the Formation of City Regions and Its Agglomeration Shadow: Evidence from the Yangtze River Delta, China"

_land, doi:10.3390/land12051053_

Round 1

Reviewer 1 Report

The Yangtze River Economic Belt is an innovation driven belt that leads China's transformation and development, as well as a coordinated development belt for interaction and cooperation between the East, West, and East of China. City-region plays a leading role in driving regional development. This paper examined the impact of state-led regional integration strategy (RIS) on city-region formation in the dimension of industrial integration and tested whether RIS causes spatial heterogeneity within the city-region. This paper has certain theoretical and practical value. It is suggested to further modify and improve the paper from the following aspects.

1. When abbreviations appear for the first time in the abstract section, the full name should be indicated;

2. County level panel data has been widely used, so it is suggested to further summarize the contribution point 2 and pay attention to the particularity of the study from the county perspective;

3. In the second part, it is suggested to add a framework diagram based on existing theoretical research on possible causal relationships, and verify the framework in the future;

4. Section 3.3 can serve as an appendix to the paper;

5. It is suggested authors supplemented the discussion section on the research results;

6. Other details, such as the labeling format of references in the text, etc.

can be further improved

Author Response

Thank you very much for your enlightening comments. I have responded point-to-point with your comments. Please see the attachment. The details are in the revised version. Thanks again for your time.

Reviewer 2 Report

Abstract

Some suggestions for improving the Abstract of the scientific article are:

·      Start the Abstract with a clear sentence that presents the research objective in a direct and concise way, such as "This paper examines the impact of state-led regional integration strategy (RIS) on city-region formation in the dimension of industrial integration and tests whether RIS causes spatial heterogeneity within the city-region."

·      In the Abstract, avoid using acronyms without explaining them first. For example, in line 13, it is better to write "using a sample of 122 county-level units in the Yangtze River Delta (YRD) region" instead of "using a sample of 122 county-level units in the YRD".

·      Try to present the study's conclusions more clearly and directly. For example, instead of "The implementation of RIS will promote the balanced layout of the secondary industry in the region, as well as the agglomeration of the tertiary industry towards the central cities", write something like "Our results show that the implementation of RIS can promote a balance in the distribution of the secondary industry in the region, with the agglomeration of the tertiary industry in the central cities".

·      Use more connectors and expressions that show the relationship between ideas, such as "however", "nevertheless", "furthermore", "consequently", etc. Avoid using long and complicated sentences. Try to break down the information into smaller and simpler sentences, making the Abstract easier to read and understand.

Congrats the author for the proper use of the DID acronym (Differences-in-Differences) and for including a case study (Jiaxing) to illustrate the results.

1 Introduction

Some possibilities for improving the introduction section of the scientific article are:

·      Starting with a broader contextualization of the topic, which can place the reader in relation to the subject and show its importance to the field of study.

·      Reinforcing the relevance of the discussion on city-regions and explaining more clearly what it means and why it is important to study.

·      Being more explicit in presenting the study objective and the problem that the research aims to solve, so that the reader knows what to expect from the article.

·      Making a smoother transition between the general discussion of city-regions and the specific presentation of the case of China, so that the change of focus does not seem abrupt.

·      Revising the writing to avoid repetitions of terms or ideas, and to make the text more fluent and easy to read.

On the other hand, the article has positive points, such as clarity in the exposition of the socioeconomic transformations related to the formation of city-regions, and the presentation of a literature review that situates the study in relation to other research already conducted on the subject.

2 Theoretical reference section

Possible improvements for the theoretical reference section of the given scientific article are:

·      Define key terms: It would be useful to clearly define terms such as "city-regions," "regional integration," "socioeconomic effect," "new regionalism approach," and "industrial integration." Clear definitions would help readers understand the author's perspective on these concepts and how they are used in the paper.

·      More literature review: The article could benefit from a more in-depth literature review that explores the existing research on city-regional integration, including different perspectives and theories.

·      Clarify research gap: The author acknowledges a gap in the literature but does not clearly articulate what that gap is. The article could benefit from a more explicit statement of the research gap, which would provide a more compelling rationale for the study.

·      Better justification for hypotheses: The article proposes two hypotheses, but it would be beneficial to provide a more detailed justification for why these hypotheses are proposed. The article could also explore alternative hypotheses that may be plausible based on existing literature.

·      Provide more context: The article would benefit from more context around the specific region of interest and how it fits into broader debates about city-regional integration.

Congrats for:

·      Good overview: The section provides a useful overview of existing literature on city-regional integration and the two dominant perspectives on the topic.

·      Clear research questions: The section clearly articulates the research questions that the paper seeks to answer.

·      Useful examples: The section provides useful examples of city-regional integration in the Yangtze River Delta and the agglomeration shadow effect.

3 Methos

The description of the study area is adequate and provides important information for contextualizing the research. The delineation of statistical units (municipalities, districts, counties, etc.) is clear and allows for a proper understanding of the sample. The explanation of the treatment (RIS) and control groups is adequate and contributes to the understanding of the analysis strategy. The choice of the analysis period (2000-2017) is adequately justified. The inclusion of Figure 1 contributes to the understanding of the sample and analysis strategy.

Possible improvements to the Methodology section of the article:

·      Provide clearer explanations of the overall research objective and the specific objectives of the Methodology section.

·      Provide more detailed criteria for selecting the cities and urban units included in the sample.

·      Clearly explain the data collection process, including sources, frequency, and the time period covered.

·      Detail the statistical methods used to analyze the data, including multivariate analysis techniques and underlying assumptions.

·      Include information about the limitations of the research and possible sources of bias.

·      Provide more detail on the data sources: The article mentions that social and economic data used in the study are at the county level, but it would be helpful to specify the exact sources of this data, such as government statistics or surveys.

·      Clarify the limitations of the methodology: While the article briefly acknowledges the limitation of classifying industry data into only secondary and tertiary sectors, it would be beneficial to discuss other potential shortcomings of the methodology, such as any assumptions or simplifications made.

·      Explain the rationale for using specific models: While the article mentions using the location quotient method and the DID and DDD models, it would be helpful to explain why these particular models were chosen and how they are expected to address the research questions.

·      Define key terms: The article uses several technical terms such as "agglomeration shadow" and "borrowed size," which may not be familiar to all readers. Including clear definitions of these terms would improve the clarity and accessibility of the methodology section.

·      Present the formulas more clearly: The formula presented for the location quotient model could be formatted more clearly and explained in greater detail to aid reader comprehension.

·      Consider adding a flowchart or diagram: A visual representation of the research process could help readers better understand the methodology and how the different steps are connected.

4 Results and Analysis  

·      Introduction of the section: It is important that the section begins with a brief introduction that explains the purpose of the presented results and how they relate to the general objectives of the article. 

·      Provide more details on the multi-stage Difference-in-Differences (DID) model used for the estimation of the total impact of RIS on industrial integration.

·      Clarify the data sources and sampling methods used in the study.

·      Provide a more detailed explanation of the terms and concepts used in the methodology, such as LQSI and LTSI.

·      Explain the rationale for choosing the multi-stage DID model and how it is better suited to address the research questions than other methods.

·      Provide more details on the robustness checks used to validate the results obtained from the model.

·      Discuss the potential limitations and assumptions of the model used, and how they may affect the validity of the results.

·      Include a section on ethical considerations, such as how the study complied with ethical guidelines for research involving human subjects and any potential conflicts of interest.

·      Provide a clear and concise summary of the methodology used, highlighting the main steps and results obtained.

·      Provide more detail on the DDD model used in the study, including the assumptions made, data sources, and the statistical techniques employed.

·      Clarify the criteria used to define small and medium-sized cities located in the vicinity of the metropolis, and provide a map or a list of these cities.

·      Provide more context on the regional integration policy and its implementation, including the timeline, the objectives, and the stakeholders involved.

·      Discuss the limitations of the DDD model, such as potential endogeneity issues, omitted variable bias, and measurement errors, and suggest ways to address these limitations.

·      Provide more detail on the industrial sectors analyzed in the study, including the criteria used to define them, the trends and challenges they face, and the policy implications of their integration.

·      Discuss the generalizability of the study's findings to other regions or countries and provide recommendations for future research in this area.

Conclusions

·      Clarify the research question and objectives in the methodology section, as well as the theoretical framework and the specific hypotheses that the study aims to test.

·      Provide a more detailed description of the data sources and data collection methods used in the study, as well as the criteria for selecting the sample of 121 urban entities in the YRD.

·      Explain in more detail the analytical methods used, specifically the DID (difference-in-difference) model and the DDD (difference-in-difference-in-differences) model, and how they were applied to the data to test the hypotheses.

·      Provide more information about the control variables used in the models, including their theoretical justification and the sources of data for these variables.

·      Provide more details on how the study evaluated spatial heterogeneity in the potential shadow area around the metropolis through the DDD model, and how this contributes to the research question and objectives.

·      Discuss the limitations of the methodology, including the data sources and the analytical methods used, and suggest directions for future research to address these limitations.

·      Provide more information on the policy implications of the research findings, including specific recommendations for local governments and policymakers, and how these recommendations relate to the theoretical framework and research question.

·      Consider the potential implications of the research findings for other regions globally seeking to promote the formation of city-regions and effect industrial restructuring, and provide more guidance on how the study's methodology and findings can be applied to other contexts.

Author Response

Thank you so much for your informative and enlightening comments! I have responded point-to-point with your comments. Please see the attachment. The details are in the revised version. Thanks again for your time.

Reviewer 3 Report

1. It is true that county-level cities are rarely studied, but it feels a bit far-fetched to use this as an important innovation point.

2. Although this article takes China as an example, it is more systematic and deeper in the introduction to explain the important role of other regions of the world;

3. The literature still needs to be updated in the latest some relevant research literature is not.

4. Lack of robustness tests;

5. The summary is not concise and does not have some policy recommendations;

Author Response

(The authors gave the same response as above.)

Round 2

Reviewer 1 Report

Suggest the author to further strengthen the discussion section, rather than mixing conclusions and discussions.

Suggest polishing the language

Author Response

Thank you so much for your informative and enlightening comments!

For point 1:The Conclusion and Discussion sections have been separated. Additionally, the exposition has been strengthened in the Discussion section.

For polishing the language: The manuscript has undergone a round of professional polishing. Thanks again!